# Boehmeria Nivea Extract (BNE-RRC) Reverses Epithelial-Mesenchymal Transition and Inhibits Anchorage-Independent Growth in Tumor Cells

**DOI:** 10.3390/ijms25179572

**Published:** 2024-09-04

**Authors:** Shiow-Ling Chen, Suh-Woan Hu, Yuh-Yih Lin, Wen-Li Liao, Jaw-Ji Yang

**Affiliations:** 1Institute of Oral Sciences, College of Oral Medicine, Chung Shan Medical University, Taichung 40201, Taiwan; slexp@csmu.edu.tw (S.-L.C.); suhwoan@csmu.edu.tw (S.-W.H.); yuhyih@csmu.edu.tw (Y.-Y.L.); aa3578852@gmail.com (W.-L.L.); 2Department of Stomatology, Chung Shan Medical University Hospital, Taichung 40201, Taiwan; 3School of Dentistry, College of Oral Medicine, Chung Shan Medical University, Taichung 40201, Taiwan

**Keywords:** Boehmeria nivea extract (BNE-RRC), epithelial–mesenchymal transition (EMT), mesenchymal–epithelial transition (MET), anchorage-independent growth, telomerase activity, reversal of EMT

## Abstract

The epithelial–mesenchymal transition (EMT) phenotype, identified as a significant clinical indicator in regard to cancer, manifests as a biological process wherein cells transition from having epithelial to mesenchymal characteristics. Physiologically, EMT plays a crucial role in tissue remodeling, promoting healing, repair, and responses to various types of tissue damage. This study investigated the impact of BNE-RRC on oral cancer cells (KB) and revealed its significant effects on cancer cell growth, migration, invasion, and the EMT. BNE-RRC induces the epithelial-like morphology in KB cells, effectively reversing the EMT to a mesenchymal–epithelial transition (MET). Extraordinarily, sustained culturing of cancer cells with BNE-RRC for 14 days maintains an epithelial status even after treatment withdrawal, suggesting that BNE-RRC is a potential therapeutic agent for cancer. These findings highlight the promise of BNE-RRC as a comprehensive therapeutic agent for cancer treatment that acts by inhibiting cancer cell growth, migration, and invasion while also orchestrating a reversal of the EMT process. In this study, we propose that BNE-RRC could be an effective agent for cancer treatment.

## 1. Introduction

The epithelial–mesenchymal transition (EMT) is a biological process that enables epithelial cells to acquire mesenchymal properties, such as increased motility, invasiveness, and resistance to apoptosis [1,2,3]. The EMT is a key process in embryonic development, tissue repair, and organogenesis, but it can also be hijacked by cancer cells to promote tumor progression and metastasis [4,5,6]. The EMT process involves the downregulation of epithelial markers, such as E-cadherin [7,8], and the upregulation of mesenchymal markers, such as vimentin and N-cadherin [9,10]. The loss of cell adhesion and the acquisition of migratory and invasive properties are characteristic of the mesenchymal phenotype.

The EMT is a complex process that is regulated by multiple signaling pathways and transcription factors. Various growth factors, cytokines, and extracellular matrix components can activate EMT signaling pathways by binding to specific receptors on the cell surface and converging signaling pathways on a set of transcription factors, including snail, slug, twist, and ZEB1/2, which regulate epithelial-related gene expression [11,12,13,14]. At the same time, these transcription factors might induce the expression of mesenchymal genes, leading to the acquisition of mesenchymal properties. In addition to the transcriptional regulation of the EMT, post-transcriptional mechanisms, such as alternative splicing, microRNA regulation [15], and protein degradation, have also been shown to play important roles in EMT regulation [16].

The EMT is also involved in the maintenance of cancer stem cells (CSCs), a subpopulation of cancer cells that possess stem-like properties and are believed to be responsible for tumor initiation, progression, and recurrence [17]. CSCs have been shown to have higher levels of EMT markers, such as snail and vimentin, and increased drug resistance compared to non-CSCs. The presence of CSCs in tumors has become an important target for cancer treatment, as they may be responsible for drug resistance and disease relapse [4,18,19].

Moreover, recent studies have expanded our understanding of the interplay between the EMT and the tumor microenvironment. This dynamic environment comprises a variety of cell types, including cancer-associated fibroblasts, immune cells, and endothelial cells, alongside extracellular matrix components and soluble factors [20,21]. Interactions between cancer cells and the tumor microenvironment trigger signaling pathways that orchestrate the EMT. These signaling cascades not only influence the phenotypic transformation of cancer cells but also play a pivotal role in the broader context of tumor development.

The contribution of the EMT to tumor progression and metastasis has rendered it an attractive target for cancer therapy. Several strategies aimed at inhibiting or reversing the EMT have been explored as potential therapeutic approaches. Small-molecule inhibitors targeting EMT-associated signaling pathways, including TGF-β, Wnt/β-catenin, and Hedgehog, have shown promise in preclinical studies [22,23,24,25,26,27,28]. These inhibitors have demonstrated the ability to moderate the EMT and impede tumor progression. However, these inhibitors have yet to be clinically approved due to their limited efficacy and potential toxicity. The balance between effectively inhibiting the EMT and maintaining normal cellular function constitutes a major obstacle in the clinical development of these inhibitors.

Natural compounds targeting EMT properties have also been identified as potential therapeutic agents. For example, curcumin, a polyphenol isolated from turmeric, has been shown to inhibit the EMT and promote the mesenchymal–epithelial transition (MET) in various cancer cell lines, leading to increased drug sensitivity. Other natural compounds, such as resveratrol, sulforaphane, and genistein, have also been reported to inhibit the EMT and promote the MET in cancer cells [29,30,31,32,33,34]. Studies suggest that MET can restore drug sensitivity by reversing the phenotypic and functional changes associated with EMT [35].

In recent years, the therapeutic potential of natural compounds derived from medicinal plants has garnered significant attention in cancer research. One such compound is Boehmeria nivea extract (BNE-RRC), also known as BNE-101, which is derived from the root extract of Boehmeria nivea, a plant traditionally used in Chinese medicine. Our previous studies demonstrated the multifaceted anti-cancer properties of BNE-RRC. Specifically, BNE-RRC has been shown to inhibit cancer cell growth, exert anti-inflammatory effects, prevent drug resistance in cancer cells, and suppress the expression of glucose-regulated protein 78 (GRP78), a key protein involved in the unfolded protein response and associated with cancer progression and chemoresistance. The anti-cancer effects of BNE-RRC are attributed to its ability to target multiple signaling pathways involved in cancer development and progression [36]. For instance, BNE-RRC has been found to downregulate the expression of cyclooxygenase-2 (COX-2), a pro-inflammatory enzyme that is often upregulated in cancer and contributes to tumor growth and metastasis. Additionally, BNE-RRC can modulate the Wnt/β-catenin signaling pathway, which plays a critical role in cancer cell proliferation and survival. By inhibiting the Akt signaling pathway, BNE-RRC induces apoptosis in cancer cells, further contributing to its anti-cancer activity. In this study, we focused on testing whether the effect of BNE-RRC on reverse epithelial–mesenchymal transition (EMT) could re-endow a chemotherapeutic drug such as Doxorubicin with sensitivity in tumor cell lines. We found that BNE-RRC could suppress the expression of EMT markers, such as vimentin and desmin, and increase the expression of the MET marker α-catenin. Thus, we suggest that BNE-RRC could play a role in inhibiting EMT and triggering MET, ultimately leading to reverse EMT and an increase in chemotherapeutic drug sensitivity. These findings suggest that BNE-RRC may be a promising therapeutic agent for the treatment of cancer, as it not only inhibits tumor growth but also enhances the efficacy of chemotherapy.

## 2. Results

### 2.1. The Effects of BNE-RRC on Cancer Cell Proliferation

The soft agar culture assay, a method in which the tumor microenvironment is mimicked by preventing cell attachment and enabling three-dimensional growth, was used to investigate the dose-dependent effects of Boehmeria nivea extract (BNE-RRC) on cancer cell proliferation. For this study, we used KB cells (ATCC CCL17), which, despite originally being thought to derive from an epidermal carcinoma of the mouth, have been shown to be a derivative of the HeLa cell line according to isoenzyme analysis, HeLa marker chromosomes, and DNA fingerprinting. This lineage means the cells do not exhibit typical epithelial characteristics but rather represent a heavily modified cancer cell model, which is important to consider when interpreting the results. KB were treated with varying concentrations of BNE-RRC (0.2, 0.5, 1, and 2 mg/mL) and monitored for their growth as colonies over 14 days. The results, illustrated in Figure 1, demonstrate a dose-dependent inhibition of cancer cell proliferation by BNE-RRC, with significant reductions in both the number and size of colonies as the concentration increased. This dose response highlights the potential of BNE-RRC as a therapeutic agent capable of effectively inhibiting cancer cell proliferation under anchorage-independent conditions.

### 2.2. The Effects of BNE-RRC Treatment on Neoplastic Cell Migration and Invasion

Most neoplastic cells exhibit high migration (M) and invasion rates (I). Cell migration plays a crucial role in many biological processes, including embryological development, tissue formation, inflammation or immune defense, and particularly cancer progression. During cancer metastasis, tumor cells disseminate from the primary site, spread through the circulatory and lymphatic systems, and subsequently colonize distant organs. Therefore, it is crucial to target both abilities to impede further dissemination. Thus, in our study, it was necessary to determine whether migration or invasion behavior was exhibited by the cancer cells subjected to treatment with BNE-RRC through the Boyden chamber membrane assay. KB cells maintained higher proliferation rates (Figure 1) under anchorage-independent conditions and exhibited migration and invasion capabilities. However, when treated with BNE-RRC, these cells lost their migration and invasion abilities (Figure 2a,b). Thus, the treatment with BNE-RRC not only suppressed migration and invasion abilities but also inhibited cancer cell proliferation in the culture of cancer cells.

Next, to further evaluate the diminished migration and invasion capacities of cancer cells subjected to treatment with BNE-RRC, we investigated whether this effect was primarily due to the lack of mobility. We tested the effect of BNE-RRC on cell migration via a wound-healing assay. Confluent monolayers of cells were scrape-wounded with a sterile plastic pipette, and the migration of cells to close the gap was monitored. The BNE-RRC-treated cells exhibited slower closure of the wound area compared to the control cells (Figure 3). Thus, these data suggest that BNE-RRC has significant potential to inhibit migration in cancer cells. Furthermore, we observed significant morphological changes in KB cells treated with BNE-RRC compared to untreated cells. In Figure 4, phase-contrast microscopic images reveal that BNE-RRC-treated cells acquire an epithelial-like morphology, suggesting the induction of mesenchymal–epithelial transition (MET). To provide further evidence of MET, Figure 5 demonstrates changes in key protein expressions associated with this transition, as assessed by Western blot analysis. Building on this biochemical evidence, we employed fluorescent microscopy to analyze the expression of vimentin, a mesenchymal marker, as detailed in Figure 6. This analysis confirms a decrease in vimentin expression, further supporting the morphological evidence of MET. Our data not only underscore the significance of BNE-RRC in inhibiting cancer cell migration but also highlight its ability to induce significant morphological changes, driving tumor cells towards an epithelial-like status.

### 2.3. BNE-RRC Reverses Epithelial–Mesenchymal Transition in Oral Cancer Cells

The above experiments indicate that BNE-RRC is effective in inhibiting the growth, migration, and invasion of cancer cells as well as inducing an epithelial morphology, indicating that BNE-RRC regulates numerous key morphological proteins associated with these changes. In order to investigate these transitional alterations in tumor cells, KB cells were cultured in the presence of BNE-RRC, and the expression of these key morphological proteins was determined. Studies have shown that cells undergoing EMT are associated with a loss of epithelial markers and the gain of expression of mesenchymal markers. In our investigation, the BNE-RRC-treated cancer cells exhibited an epithelial morphology, suggesting a reversal from epithelial–mesenchymal transition to mesenchymal–epithelial transition (MET). To further support this finding, we examined the expression of the epithelial protein α-catenin and found that BNE-RRC significantly induced α-catenin protein expression in the cancer cells treated with this compound (Figure 5). This experiment suggests that BNE-RRC can effectively trigger the epithelial phenotype in cancer cells. On the other hand, we examined the expression of the mesenchymal markers desmin and vimentin and found that the levels of both proteins were suppressed in the BNE-RRC-treated cancer cells (Figure 5), indicating the loss of MET in these cancer cells under the treatment with BNE-RRC. These results suggest that BNE-RRC has the potential to reverse EMT and induce MET in cancer cells, showing that it could be a promising therapeutic agent for cancer treatment.

**Figure 5 ijms-25-09572-f005:**
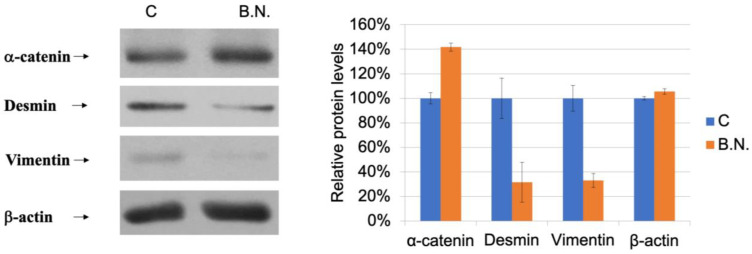
Induction of MET in cancer cells by BNE-RRC. KB cells were treated with BNE-RRC (2 mg/mL), and Western blot analysis was performed to detect the expression of key proteins involved in the EMT/MET process. The BNE-RRC-treated cancer cells exhibited an epithelial morphology, suggesting a reversal of EMT to MET. The expression of the epithelial protein α-catenin was significantly induced in the BNE-RRC-treated cancer cells, while the expression of the mesenchymal markers desmin and vimentin was downregulated. The relative protein expression levels are shown in the right panel. These results suggest that BNE-RRC has the potential to reverse EMT and induce MET in cancer cells. These experiments were performed five times; representative data are shown.

### 2.4. BNE-RRC Induces and Maintains Epithelial Status in Cancer Cells by Inhibiting the Mesenchymal Phenotype

Post-treatment with BNE-RRC, KB cells demonstrated a notable decrease in the mesenchymal marker vimentin, indicating a shift towards an epithelial phenotype (Figure 6a,b). Even after the withdrawal of BNE-RRC treatment, these effects persisted, as the cells continued to display an epithelial status, designated as KB B.N.R (Figure 6a,b). These results underscore the potential of BNE-RRC to induce sustainable changes in cell morphology indicative of MET. Our results suggest that BNE-RRC treatment can effectively induce and maintain an epithelial status in cancer cells, offering a promising therapeutic strategy for cancer treatment. Moreover, our findings demonstrate that BNE-RRC not only induces the reversal of EMT but also maintains an epithelial phenotype in oral cancer cells for an extended period, even after the withdrawal of the treatment. This sustained effect is a novel observation that highlights the potential of BNE-RRC as a long-term therapeutic agent for inhibiting EMT in cancer.

**Figure 6 ijms-25-09572-f006:**
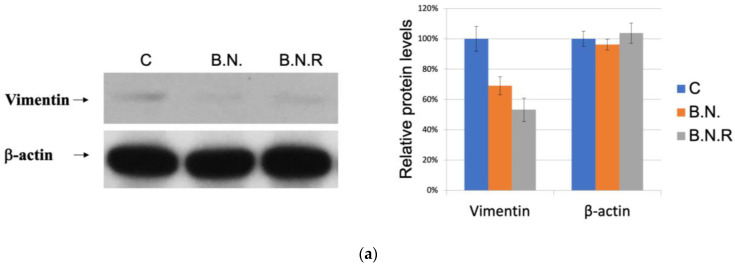
The effects of BNE-RRC treatment on the epithelial phenotype in cancer cells. (**a**) The results of a Western blot analysis showing lower expression of the mesenchymal marker vimentin in KB cells treated with BNE-RRC for 14 days (B.N.) compared to the untreated cells. After the cessation of the BNE-RRC treatment, the cancer cells maintained low levels of vimentin expression for an additional 4 weeks (B.N.R.), indicating a lasting effect of the BNE-RRC treatment on inducing an epithelial morphology. The relative protein expression levels are shown in the right panel. (**b**) The results of an immunofluorescence analysis of cancer cells treated with BNE-RRC for 14 days (B.N.) and then cultured for an additional 4 weeks without BNE-RRC treatment (B.N.R.). The BNE-RRC-treated cells exhibit low levels of vimentin. These experiments were performed three times; representative data are shown.

### 2.5. Induction of the Epithelial State by BNE-RRC Reverses EMT and Increases Sensitivity to Chemotherapy in Cancer Cells

In this study, we investigated the impact of BNE-RRC treatment on cancer cells, specifically focusing on its potential to induce a reversal from epithelial–mesenchymal transition (EMT) to mesenchymal–epithelial transition (MET) status. The rationale for this investigation is based on the crucial role of EMT in increasing malignancy, including via enhancing migration, invasion, and resistance to therapeutic interventions. Our data reveal that BNE-RRC treatment not only induced a reversal from a mesenchymal to an epithelial status in cancer cells but also maintained this altered phenotype even after an extended period in a BNE-RRC-free medium. This observation suggests that the effects of BNE-RRC treatment are long-lasting and could serve as a promising therapeutic strategy for cancer treatment.

To understand the clinical consequences of the observed transition, we examined the responsiveness of the BNE-RRC-treated cells to the chemotherapeutic drug doxorubicin. We found that the reversion to an epithelial status rendered these cells more sensitive than the parental cells (KB) to doxorubicin, as demonstrated in Figure 7. This result indicates that the induction of MET by BNE-RRC increases the sensitivity of cancer cells to chemotherapeutic agents, suggesting that the combination of BNE-RRC treatment with chemotherapy may overcome therapeutic resistance. Furthermore, our findings reinforce the idea that mesenchymal cells are generally more resistant to chemotherapeutic drugs than epithelial cells. The BNE-RRC treatment’s ability to reverse the EMT state highlights its potential as a novel therapeutic approach for sensitizing cancer cells to chemotherapy.

Our study suggests that BNE-RRC may modulate key signaling pathways involved in EMT in a manner distinct from that of other natural compounds previously reported. This provides novel mechanistic insights into the anti-EMT effects of BNE-RRC and suggests potential targets for future therapeutic intervention.

### 2.6. Exploring the Effects of BNE-RRC Treatment on Telomerase Activity in Cancer Cells

Telomeres play a critical role in maintaining the genomic integrity of normal cells, preventing chromosomal instability and ensuring the faithful transmission of genetic material from parental cells to daughter cells. However, in the majority of cancer cells, telomere length is maintained, allowing the cancer cells to continue dividing and surviving. Due to the importance of telomere length and telomerase activity in cancer initiation and progression, we aimed to investigate the effect of BNE-RRC treatment on telomerase activity in cancer cells. We treated cancer cells with BNE-RRC for a period of 14 days and then extracted the BNE-RRC to assess whether telomerase activity remained suppressed in both the B.N. and B.N.R. groups. Our results demonstrate that the BNE-RRC treatment was able to suppress telomerase activity in the cancer cells, even after the long-term culturing of cells in the absence of BNE-RRC (Figure 8). This finding suggests that BNE-RRC treatment could impair the ability of cancer cells to maintain telomere length and telomerase activity, thereby limiting their ability to divide and survive. Furthermore, the long-term suppression of telomerase activity observed in our study suggests that the effects of BNE-RRC treatment might allow it to serve as a long-lasting therapeutic agent for cancer.

### 2.7. Identification of Compounds through HPLC Analysis

According to the existing literature, Boehmeria nivea plants contain several prominent compounds, including isoquercitrin, p-coumaric acid, caffeic acid, and rutin [37]. The extract used in this study was supplied by CAXXION Corp, which adheres to standardized extraction methods ensuring consistency with those used in prior studies by Yang J-J [36]. In our study, high-performance liquid chromatography (HPLC) was employed to identify and separate key compounds in the BNE-RRC. This analysis resulted in the separation of p-coumaric acid (1), isoquercitrin (2), and caffeic acid (3), with retention times of 12.1, 17.4, and 19.1 min, respectively (Figure 9). Our results confirm high levels of isoquercitrin and p-coumaric acid, consistent with previous findings. However, numerous unidentified compounds are also present, necessitating further research to characterize these components and assess their roles. Additionally, mass spectrometry analysis, similar to that conducted by Yang J-J in 2018, was employed to further verify the identities and purities of the detected compounds, ensuring rigorous scientific validation of the BNE-RRC used in our experiments.

## 3. Discussion

Plants and botanicals serve as enormous sources of natural compounds that have potential therapeutic applications. One such compound is BNE-RRC, a root extract from Boehmeria nivea, which has been discovered to inhibit cancer cell growth. Previous studies have shown that the treatment of cancer cells with BNE-RRC results in the inhibition of various proteins associated with cancer development and survival. In addition, BNE-RRC has been found to reverse epithelial–mesenchymal transition (EMT) in cancer cells, a process known to play an important role in tumor progression and metastasis. EMT is a cellular phenomenon wherein epithelial cells lose their characteristic morphology and acquire mesenchymal properties, leading to increased cell motility, invasiveness, and resistance to apoptosis. EMT is a complex process regulated through the activation of EMT-inducing transcription factors (EMT-TFs), including zinc-finger E-box-binding homeobox factors such as SNAIL, ZEB1, SLUG, and the basic helix–loop–helix factors TWIST1 and TWIST2 [38,39,40]. The expression of these EMT-TFs is tightly regulated by various signaling pathways, non-coding RNAs, and extracellular mediators [41,42,43]. While EMT enhances cancer cell survival and metastatic potential, inducing MET through therapeutic agents can reverse these effects and is seen as a promising strategy to enhance drug efficacy. MET not only reverts the cellular phenotype but also decreases malignancy and increases susceptibility to chemotherapeutic agents [44]. EMT is classified into three types based on the biological context in which it occurs. Type I EMT is involved in embryonic development, type II EMT is observed during tissue regeneration and wound healing, and type III EMT occurs during carcinoma progression. In cancer, EMT is associated with the acquisition of stem-cell-like properties, increased resistance to chemotherapy and immunotherapy, and the ability to form distant metastases [45,46,47]. Several studies have shown that blocking EMT or reversing EMT to induce mesenchymal–epithelial transition (MET) can enhance the efficacy of chemotherapy and immunotherapy in cancer [45,48]. MET is a cellular process that is the opposite of EMT, where mesenchymal cells transition back to their epithelial phenotype, resulting in increased cell–cell adhesion, decreased cell motility, and increased sensitivity to apoptosis. Therefore, understanding the regulatory mechanisms of MET and its impact on drug resistance could significantly enhance the design of novel therapeutic strategies aimed at modulating this transition to overcome drug resistance in cancer therapy.

A previous study showed that BNE-RRC inhibits cancer cell growth through various mechanisms, including the suppression of inflammatory proteins, like COX2; the regulation of the cancer development regulation protein β-catenin; the inhibition of the cancer cell survival protein AKT; the downregulation of the anti-apoptotic protein GRP78; and the reduction in the invasion-associated protein MMP9 [36]. In addition to its anti-cancer properties, BNE-RRC can reverse the EMT in cancer cells by suppressing mesenchymal markers, such as desmin and vimentin, and increasing the expression of the MET marker α-catenin. This reversal enhances the sensitivity of cancer cells to chemotherapeutic drugs, particularly doxorubicin. This study also shows that the prolonged administration of BNE-RRC to cancer cells maintained the MET characteristics of the treated cells for over four weeks, leading to a gain of chemotherapeutic drug sensitivity. This study suggests that converting EMT to MET contributes to the efficacy of the treatment of cancer with chemotherapeutic drugs. The study presented here demonstrates the potential of BNE-RRC as a promising therapeutic candidate for cancer treatment. Its ability to inhibit cancer cell growth and reverse EMT, leading to an epithelial phenotype in cancer cells, highlights its significance, as the reversal of the EMT contributes to increased sensitivity to chemotherapeutic drugs, a crucial aspect in the treatment of cancer. 

Further research is needed to unravel the mechanism by which BNE-RRC triggers the discussed morphological transition and the impact of this transition on cancer cells. Nevertheless, the promising results of this study suggest that BNE-RRC has significant potential as a therapeutic candidate for cancer treatment. In addition, the results of this study may have stronger implications for the development of cancer therapies through targeting the EMT. The EMT plays a significant role in cancer progression, metastasis, and the development of therapeutic resistance. Understanding the mechanisms regulating this process and developing therapies that can reverse it could enhance cancer treatment and reduce the burden on patients. Overall, the study presented herein provides important insights into the potential of BNE-RRC as a therapeutic candidate for cancer treatment. BNE-RRC has the potential to improve cancer treatment outcomes by targeting the EMT process and inducing EMT reversal. Further research is needed to fully elucidate BNE-RRC’s mechanisms of action and develop more effective and targeted therapies for cancer treatment. This study provides crucial insights into BNE-RRC’s therapeutic potential, especially its capacity to sustain epithelial characteristics long after treatment cessation, thereby supporting its role in stable MET induction. Such enduring effects suggest that BNE-RRC not only triggers MET but also maintains this state, potentially increasing the effectiveness of treatments that target cellular morphology and behavior. These findings highlight the need for further investigation into how BNE-RRC enhances drug sensitivity through MET, which could lead to more targeted and effective cancer therapies.

The plant kingdom is an abundant source of numerous natural compounds for human health research. In previously study, we employed BNE-RRC to effectively inhibit cancer cell growth by regulating key proteins associated with cancer development, growth, and survival, such as COX2, AKT, GRP78, and MMP9. Additionally, we observed a reversal of the EMT to an epithelial phenotype in cancer cells. EMT has significant consequences for clinical oncology, as it enhances tumor cell resistance to chemotherapy and immunotherapy. It plays an important role in tumor progression and metastasis, making this process a target for cancer therapy. In the runup to the EMT, the expression of cellular markers such as E-cadherin and α-catenin is suppressed, while the expression of mesenchymal markers, including N-cadherin, vimentin, and desmin, is induced [7,8,49,50]. The blockage or reversal of EMT can restore the sensitivity of tumor cells to several therapeutic regimens [44,51,52]. This study found that BNE-RRC suppresses the expression of mesenchymal markers such as vimentin and desmin and increases the expression of the MET marker α-catenin, thereby reversing EMT. EMT processes are classified into three types based on the biological context: type I EMT is involved in embryonic development, type II EMT is observed during tissue regeneration and wound healing, and type III EMT occurs during carcinoma progression. EMT processes can take place in normal tissues and neoplastic growths, depending on which EMT-TFs are expressed. EMT-TFs include SNAIL, ZEB1, SLUG, TWIST1, and TWIST2 [15,23,40,53]. These EMT-TFs are tightly regulated via oncogenic signaling, non-coding RNAs, and extracellular molecules. It remains to be established whether BNE-RRC treatment regulates the expression of these key EMT-TFs, controlling both EMT and the reverse process, MET.

In this study, we investigated the composition of BNE-RRC using HPLC and identified several compounds, including p-coumaric acid, isoquercitrin, and caffeic acid. Although the anti-cancer effects of BNE-RRC might involve a mixture of compounds, the individual effects of these compounds need to be further investigated. P-coumaric acid, a phenolic acid found in BNE-RRC, has been reported to exhibit anti-cancer properties in various studies. For instance, it has been shown to induce apoptosis and inhibit proliferation in cancer. Our ongoing research aims to compare the anti-cancer effects of p-coumaric acid with those of BNE-RRC to elucidate the former’s role in the extract’s overall activity. Isoquercitrin, a flavonoid also present in BNE-RRC, has been reported to engage in anti-inflammatory and anti-cancer activities. Caffeic acid, another phenolic compound identified in BNE-RRC, has been the subject of debate regarding its anti-cancer effects. While some studies suggest that caffeic acid may exhibit anti-cancer activity, others argue that its effects may be context-dependent and vary across different types of cancer cells. HPLC analysis of BNE-RRC compounds provides valuable insights into the potential bioactive components contributing to its anti-cancer effects. Further studies are needed to elucidate the individual roles of these compounds and their synergistic interactions within this extract.

The findings from our study provide compelling evidence that Boehmeria nivea extract (BNE-RRC) effectively inhibits epithelial–mesenchymal transition (EMT) and promotes mesenchymal–epithelial transition (MET) in cancer cells. These transitions are critical in cancer progression and metastasis, and their modulation by BNE-RRC could represent a novel therapeutic strategy to enhance drug sensitivity and reduce therapeutic resistance. Our study highlights that BNE-RRC impacts key molecular pathways that regulate EMT and MET. The extract downregulates the expression of transcription factors such as snail and slug, which are pivotal in promoting EMT by repressing epithelial marker genes and activating mesenchymal genes. By inhibiting these factors, BNE-RRC prevents the loss of epithelial characteristics and maintains cellular adhesion, which is often compromised during cancer progression. 

The use of KB cells, which are known to be contaminated with HeLa cells, raises considerations regarding the typical EMT characteristics. While these cells do not exhibit classical epithelial traits, their robust mesenchymal features make them a valuable model for studying the reversal effects of MET. Despite this, we acknowledge that our findings should be interpreted with caution and validated in additional cell lines with well-defined epithelial properties to ensure the generalizability of our results across different cellular contexts.

The inhibitory effect of BNE-RRC on key transcription factors like snail and slug is crucial, as these proteins directly regulate genes responsible for cell adhesion and motility. Downregulation of these factors leads to an upregulation of epithelial markers and a suppression of mesenchymal markers, facilitating a switch back to an epithelial phenotype that is typically more sensitive to chemotherapeutic agents. This sensitivity is partly due to the decreased expression of proteins such as P-glycoprotein, a drug efflux transporter often upregulated in mesenchymal cells. Future studies should focus on quantifying these changes in drug efflux capacities and exploring the effect of BNE-RRC on other aspects of drug metabolism.

Our findings provide valuable insight into the sustained effects of BNE-RRC treatment, which can reverse the EMT and maintain the MET phenotype for over four weeks, contributing to an increased sensitivity to chemotherapeutic drugs. The EMT process is known for generating neoplastic stem cells and elevating therapeutic resistance; BNE-RRC reverses this program, providing a potential avenue for cancer therapy. The conversion of EMT to MET by BNE-RRC appears to establish a counter-regulatory mechanism against cancer development and metastasis. This transition contributes to the effectiveness of chemotherapeutic drugs in cancer patients. BNE-RRC suppresses typical mesenchymal markers, such as desmin and vimentin, suggesting a unique role of BNE-RRC in regulating MET. Our results suggest that BNE-RRC induces morphological changes in tumors, reducing the mesenchymal stage and increasing tumors’ susceptibility to various treatments, including chemotherapies. However, the precise mechanism by which BNE-RRC triggers this transition requires further investigation.

## 4. Materials and Methods

### 4.1. Cell Culture

The nasopharyngeal carcinoma cell line KB CRL-3596 (ATCC CCL-17) was maintained at 37 °C in a humidified environment containing 5% CO_2_. The cells were grown in Dulbecco’s Modified Eagle medium (DMEM) (supplied by Invitrogen Corporation, Carlsbad, CA, USA) enriched with 10% fetal bovine serum (FBS) and antibiotics (penicillin and streptomycin, each used in concentrations of 25 U/mL). To develop Boehmeria nivea extract-resistant (BNR) cells, KB cells were treated with 2 mg/mL of Boehmeria nivea extract (BNE-RRC) over an extended period. Following this treatment, cells were cultured in BNE-RRC-free DMEM to study the persistence of induced changes. These cells are referred to as “BNR” cells throughout the study. To evaluate the durability of the mesenchymal-to-epithelial transition (MET) induced by BNE-RRC, KB cells were initially treated with BNE-RRC for 14 days. Subsequently, these cells were transferred to fresh DMEM without BNE-RRC and maintained for an additional four weeks. This experimental setup was designed to observe the long-term effects of BNE-RRC on cellular morphology and phenotype stability.

### 4.2. Soft Agar Assay

The cells were resuspended in DMEM containing 10% FBS and 0.3% Agar Noble (BD, Difco, MD, USA) and then seeded onto a base layer comprising 0.5% Agar Noble. This assay was conducted in 6-well plates, seeded with approximately 2500 to 10,000 cells per well, with each condition replicated three times. After 2 weeks of incubation at 37 °C and with 5% CO_2_, the colonies were stained with Iodonitrotetrazolium chloride (INT) solution, photographed, and manually counted. The quantification of colonies was performed using an inverted light microscope at a magnification of 40×. The results are presented as the average number of colonies ± standard error (SE) calculated from six fields across three independent wells.

### 4.3. Western Blot Analysis

For protein extraction, cells were lysed in a buffer composed of 50 mM Tris-HCl (pH 8.0), 150 mM NaCl, 1% NP-40, 1 mM EDTA, supplemented with protease inhibitors (1 mM PMSF, 10 ng/mL leupeptin, 50 mM NaF, and 1 mM sodium orthovanadate). Proteins were separated on a 10% SDS-PAGE gel and transferred to a PVDF membrane. The membrane was blocked with 5% non-fat milk in TBS-T for 1 h at room temperature, then incubated with primary antibodies overnight at 4 °C. After washing, the membrane was incubated with HRP-conjugated secondary antibodies for 1 h at room temperature. The blots were developed using an enhanced chemiluminescence detection system. Specific antibodies used were anti-α-catenin, anti-vimentin, and anti-desmin. Western blot analysis was routinely performed to determine the expression levels of key proteins involved in the EMT/MET process. Each protein was analyzed across a minimum of five independent experiments to ensure the reliability of the data. The blots were developed using an enhanced chemiluminescence detection system, and the bands were quantified using densitometry. The representative blots shown were selected based on their clear demonstration of the typical results obtained.

### 4.4. Wound-Healing Assay

KB cells were plated on 10 cm dishes and grown to confluence. A scratch was made across the cell monolayer using the tip of a 200 µL pipette. The cells were incubated in DMEM containing 10% FBS, and images were captured 18 h post-scratch using a Axiovert 200 microscope (Carl Zeiss AG, Oberkochen, Germany). The images were analyzed using Image-Pro software (version 6.0, Media Cybernetics, Inc., Rockville, MD, USA) to determine the percent of gap closure. The area of the initial scratch (0 h) and the area remaining unclosed at 18 h were measured, and gap closure was calculated as a percentage reduction in the original wound area. Each condition was performed in triplicate, and the experiment was repeated three times to ensure reproducibility.

### 4.5. Cell Migration Assay

A Boyden chamber migration assay was conducted by adding DMEM supplemented with 10% FBS to the lower compartment of the chamber, equipped with polyvinylpyrrolidone-free polycarbonate membranes featuring 8 µm pores (Neuro Probes, Inc. Gaithesburg, MD, USA). To the upper compartment, 1500 cells per well were seeded in serum-free DMEM. After incubation for 24 h at 37 °C, facilitating cell migration through the membrane, the membranes were stained with Giemsa. Migrated cells in the lower chamber were quantified using a counting grid inserted into the eyepiece of a phase-contrast microscope at 40× magnification, covering nine fields per well to ensure representative sampling. Enhanced imaging techniques were applied to improve visualization and accuracy of cell counting.

### 4.6. Telomerase Activity Assay

Telomerase activity was assessed using the TRAPEZE^®^ Gel-Based Telomerase Detection Kit (Sigma-Aldrich, St. Louis, MI, USA). Cell extracts were prepared according to the kit’s instructions. Briefly, the KB cells were pelleted and resuspended in 1× CHAPS Lysis Buffer, and the protein concentration was determined. Tissue samples were homogenized in 1× CHAPS Lysis Buffer. The homogenates were then centrifuged at 12,000× *g* for 20 min at 4 °C, and the supernatants were collected. For the telomerase activity assay, aliquots of the supernatant were quick-frozen on dry ice and stored at −85 °C to −75 °C. The extracts were aliquoted into small volumes to prevent frequent freeze–thaw cycles. Each assay was performed with 2 µL of cell extract, corresponding to 0.5 µg of total protein for cell extracts and 10–500 ng/µL for tissue extracts. Telomerase activity was evaluated by incubating the samples at 85 °C for 10 min to inactivate telomerase, followed by PCR amplification.

### 4.7. HPLC Conditions 

This analysis was performed using an Amersham HPLC system equipped with a reverse-phase INNO C18 column (250 × 4.6 mm, 5 μm). The volume injected was 100 μL, and a DAD-WR detector was used for the analyses. The column’s temperature was consistently maintained at 25 °C, and the flow rate was adjusted to 1 mL/min. Detection occurred at a wavelength of 240 nm, employing a gradient elution technique with a binary mobile phase. The composition of the mobile phase included (A) 0.1% trifluoroacetic acid in water and (B) 0.1% trifluoroacetic acid in 80% acetonitrile (ACN). The gradient program commenced with 95% A, holding for 0 s (immediate start), and then a linear transition to 100% B over 40 min. Afterward, the system was returned to 0% B within 5 min and maintained for 60 min before reverting to the initial condition of 95% A. All the samples were analyzed in triplicate.

## 5. Conclusions

Our investigation reveals that a 14-day treatment of tumor cells with BNE-RRC led to a significant decrease in the expression of the mesenchymal marker vimentin. Interestingly, even after a 4-week withdrawal of BNE-RRC from the culture, the epithelial phenotype was sustained, rendering the tumor cells more susceptible to the chemotherapeutic drug doxorubicin. These findings underline the potential of BNE-RRC to induce a reversal of EMT to MET, thus increasing susceptibility to chemotherapeutic drugs.

## Figures and Tables

**Figure 1 ijms-25-09572-f001:**
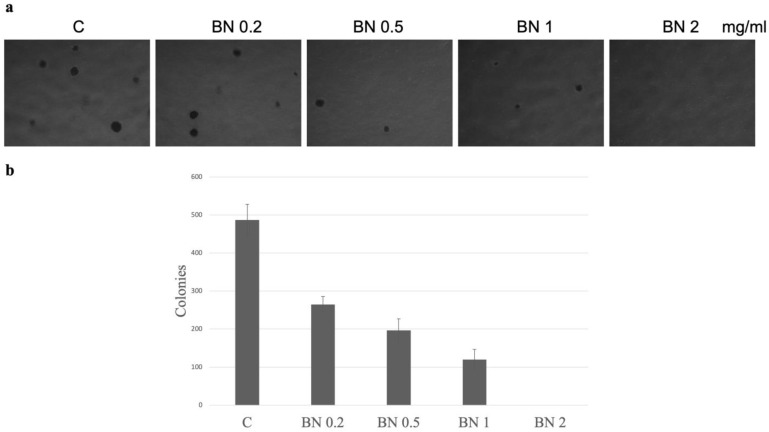
Dose-dependent inhibition of oral cancer growth by BNE-RRC in a soft agar culture assay. (**a**) Phase-contrast images showing the proliferation of oral cancer cells (KB) under anchorage-independent condition after 14 days of treatment with increasing concentrations of BNE-RRC (0.2, 0.5, 1, and 2 mg/mL), followed by INT staining. Higher concentrations of BNE-RRC progressively reduced the number and size of cancer cell colonies. (**b**) Graphical representation quantifying the number of colonies at each BNE-RRC concentration, confirming the dose-dependent inhibitory effect. ‘C’ denotes the control group, while ‘BN 0.2’, ‘BN 0.5’, ‘BN 1’, and ‘BN 2’ represent groups treated with 0.2, 0.5, 1, and 2 mg/mL of BNE-RRC, respectively. Experiments were repeated six times, with representative data shown.

**Figure 2 ijms-25-09572-f002:**
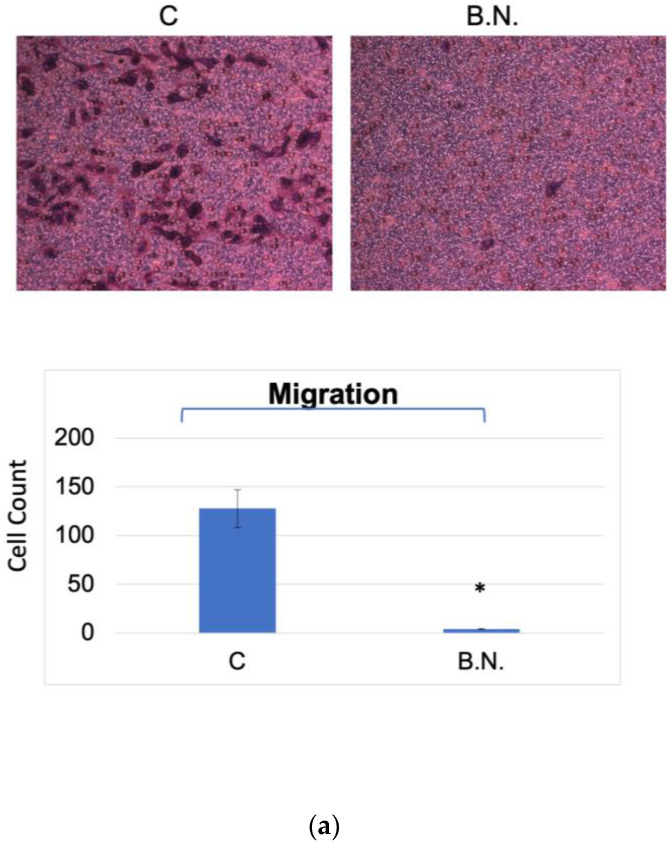
Inhibition of migration and invasion capacity in cancer cells by BNE-RRC. An evaluation of the migration and invasion capacity of cancer cells subjected to treatment with BNE-RRC through the Boyden chamber membrane assay is shown. The treatment of BNE-RRC (2 mg/mL) significantly suppressed migration (**a**) and invasion (**b**) ability in KB cells. Statistical analysis was performed using a Student’s *t*-test. The *p*-value for migration was 0.0030 (*), indicating a significant difference between the control and BNE-RRC-treated groups. For invasion, the *p*-value was 0.0010 (*), also indicating a significant difference. These results demonstrate that BNE-RRC effectively inhibited both the migration and invasion capacities of KB cells. These experiments were performed three times; representative data are shown.

**Figure 3 ijms-25-09572-f003:**
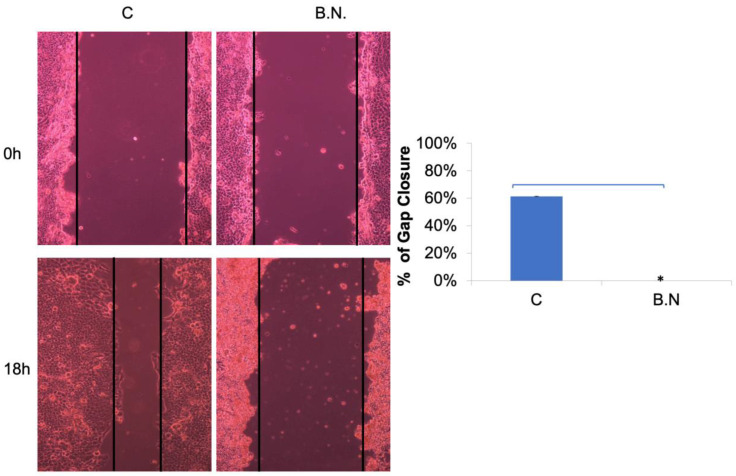
Inhibition of cancer cell migration by BNE-RRC. A wound-healing assay was employed to evaluate the effect of BNE-RRC on cell migration. Confluent cancer cells were scraped with a sterile plastic pipette with or without BNE-RRC (2 mg/mL) treatment. In the left panel, representative photos show the wound-healing progress after 18 h for the control cells and the cells treated with BNE-RRC (B.N.). The right panel presents the percentage of gap closure; the control cells achieved 61% gap closure, whereas the B.N.-treated cells exhibited no gap closure after 18 h of treatment. These data were acquired from 10 fields for each condition. Statistical analysis was performed using a Student’s *t*-test. The *p*-value for gap closure was 0.0010 (*), indicating a significant difference between the control and BNE-RRC-treated groups. These experiments were performed four times; representative data are shown.

**Figure 4 ijms-25-09572-f004:**
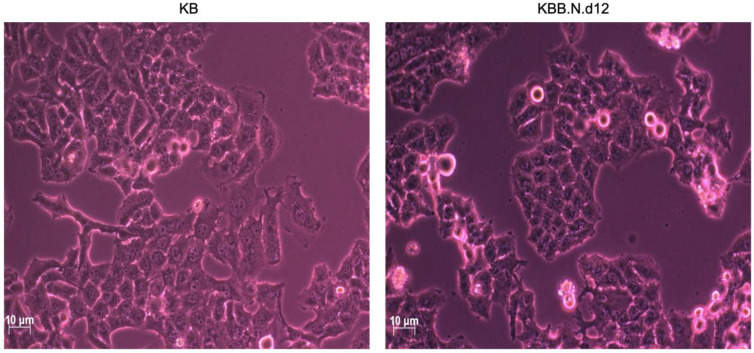
Induction of morphological changes in KB cells by BNE-RRC. The effect of BNE-RRC on cell morphology was evaluated through phase-contrast microscopy. The untreated control KB cells (**left panel**) exhibited a mesenchymal-like morphology with non-uniform cell shapes. In contrast, the KB cells treated with BNE-RRC (2 mg/mL) for 12 days (**right panel**) acquired a more rigid and uniform epithelial-like morphology.

**Figure 7 ijms-25-09572-f007:**
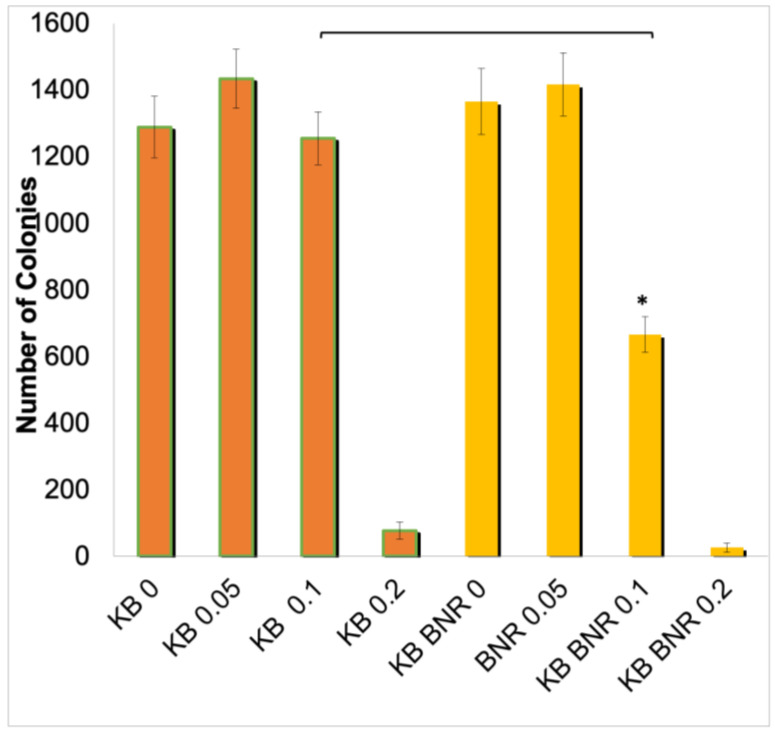
Dose-response effect of doxorubicin on KB and KB B.N.R cells treated with BNE-RRC. The graph compares the number of colonies formed in control KB cells and BNE-RRC treated KB B.N.R cells at 0.1 µM doxorubicin. Control cells formed approximately 1288 colonies, with a slight reduction to 1254 colonies upon doxorubicin treatment. KB B.N.R cells showed a reduction from 1365 colonies (control) to 665 colonies with doxorubicin, highlighting increased drug sensitivity. Data were analyzed using a Student’s *t*-test; significance marked by asterisks (* *p* < 0.005). The experiment was replicated four times.

**Figure 8 ijms-25-09572-f008:**
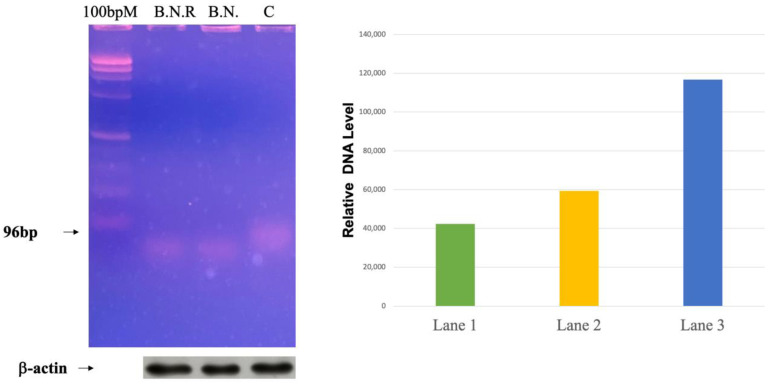
Long-term effects of BNE-RRC on telomerase activity in cancer cells. Following a 14-day treatment with BNE-RRC, cells were cultured in drug-free medium to assess persistence of telomerase suppression. Results show a significant long-term decrease in telomerase activity, indicated by reduced telomere levels in the right panel. Western blot analysis confirms equal protein loading, with β-actin as the loading control. This experiment was conducted four times.

**Figure 9 ijms-25-09572-f009:**
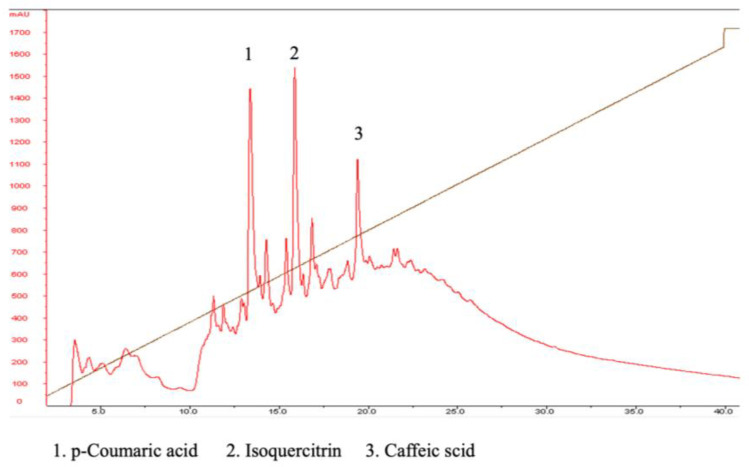
HPLC chromatographic analysis of BNE-RRC compounds. An HPLC analysis was used to identify and separate key compounds present in Boehmeria nivea extract (BNE-RRC). This analysis separated p-coumaric acid (1), isoquercitrin (2), and caffeic acid (3), with retention times of 12.1, 17.4, and 19.1 min, respectively. These experiments were performed three times; representative data are shown.

## Data Availability

No new data were created or analyzed in this study. Data sharing is not applicable to this article.

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
