# Peer review of "Boehmeria Nivea Extract (BNE-RRC) Reverses Epithelial-Mesenchymal Transition and Inhibits Anchorage-Independent Growth in Tumor Cells"

_ijms, 2024, doi:10.3390/ijms25179572_

Round 1

Reviewer 1 Report

Comments and Suggestions for Authors

Dear Authors,

Thank you for submitting your manuscript entitled "Boehmeria Nivea Extract (BNE-RRC) Reverses Epithelial-Mesenchymal Transition and Inhibits Anchorage-Independent Growth in Tumor Cells." This work builds upon a previously published study that explored the effects of BNE on the molecular mechanisms associated with inhibiting tumor cell growth. The current study delves into the impact of BNE-RRC on KB cells, revealing its ability to induce an epithelial-like morphology, inhibit cancer cell growth, migration, and invasion. The findings suggest that BNE-RRC has the potential to be a therapeutic candidate for cancer treatment. However, further research is necessary to elucidate the mechanisms of action and to develop targeted therapies.

The experiments designed and carried out to demonstrate the effects were well-planned and executed. However, the descriptions of the materials, results, figure legends, and the discussion section all necessitate thorough revision and refinement to ensure accuracy and clarity.

  1.  The use of KB cells (ATCC CCL17), which has been demonstrated to be a derivative of the HeLa cell line, raises significant concerns. According to the ATCC website, "This line was originally thought to be derived from an epidermal carcinoma of the mouth, but was subsequently found, based on isoenzyme analysis, HeLa marker chromosomes, and DNA fingerprinting, to have been established via HeLa cell contamination." The authors should acknowledge that this cell line does not exhibit the characteristics of epithelial cells, but rather represents a heavily modified cancer cell model.  
  2. In lines 132-136, Figure 4, which presents Phase-contrast microscopic images of cells treated and untreated with BNE, indicating a change in epithelial-like morphology to mesenchymal. To enhance the credibility of this section, image analysis should be conducted to demonstrate changes in morphological parameters. Authors can utilize open-source software like ImageJ, Cell Profiler, or commercially available software such as SignalI Image Artist, MATLAB, or Stratominer to derive values from image analysis. Correct controls demonstrating the "correct" morphology of epithelial and mesenchymal cells should be included. Alternatively, the authors could use fluorescent microscopy and detect specific biomarkers for each type of cell, as mentioned in the current work.
  3. Additionally, it is important to incorporate more information about Mesenchymal-Epithelial Transition (MET) as a process that can contribute to drug sensitivity in cancer. The introduction (lines 63-65) and discussion (paragraphs 287-298) lack information and references specifically addressing MET and its potential for control as a process to sensitize cancer cells to drug treatment.
  4. The Boehmeria Nivea Extract is obtained from the plant and is a natural product containing a mixture of chemical molecules. However, the authors have not provided details about the extract's source, preparation methods, or its similarity to the extract used in previously published works by Yang J-J , 2018. Without this information, the scientific validity of the entire work is compromised. Furthermore, the HPLC chromatogram presented in the study identifies specific compounds without providing results for individual reference compounds or information about Mass Spectrometry analysis, which was conducted in the previous work by Yang J-J in 2018. As a result, section lines 252-266 lacks scientific identification of the components of the extract.

Other issues and suggestions:

  1. There is issue with the inconsistent use of acronyms and full names for EMT, MET, and BNE throughout the manuscript. It is recommended to initially present the full name together with its acronym and then maintain consistency by using the acronyms without repeating combinations of both throughout the manuscript. This approach will enhance consistency in the use of terminology.
  2. List of the problems including lack of correct figure legends, incorrect organization of the Figure descriptions and results sections, lack of information in Material and Methods:

6.1.The organization of Figure 1 and its legends requires improvement. It is suggested that the figure be divided into two labeled parts: A for the images and B for the graph. The images in Part A need better contrast, and a significant portion of the figure legend should be relocated to the Results or Discussion section.

6.2.      In the Results section of Figure 1, it is essential to provide a detailed explanation of the statistics used, including the sample size (n), the average (ave), and the standard deviation (SD). Additionally, at 40x magnification, six fields per well may cover a small area and may not be representative of the actual colony counts. Increasing the number of fields per well is recommended. Furthermore, scale bars should be added to each image. If each condition was tested in only one well, the statistical significance may be compromised due to a low sample size (n=1), and it should be increased to at least three wells per condition.

6.3.      In the section titled "The effect of BNERRC treatment,…"

6.4.       Lines 106-109it is recommended to move the majority of the content into the discussion section, leaving only the description of the test and results. This section should discuss the observed values and results presented in Figure 2. A subdivision for the correct legend and references in the results is needed, and scale bars should be added to the cell images. Both graphs in Figure 2 need to have the Y-axis labeled.

6.5.      Regarding the "cell migration" method section, specifically lines 392-394, it is important that the cell counting was conducted manually using an eyepiece with a grid. The explanation should include details about the number of fields per well used for analysis and the size of the field. Additionally, the poor quality of the images and the method of cell counting using an eyepiece make it difficult to understand the accuracy of the results. More evidence supporting the accuracy of these results would be very helpful.

6.6.      Regarding Figure 3 and the "wound healing assay" method section (lines 383-386), it is essential to explain in the methods how the images were analyzed to derive the percent of Gap Closure. A detailed description of the image analysis method, including the software or techniques used, is necessary for reproducibility and understanding the accuracy of the results. Furthermore, it is important to provide statistics on how many wells were treated and used for analysis to ensure the reproducibility of the results. This information will contribute to the transparency and reliability of the experimental data.

6.7.       In regards to Figure 5 and the description of the Western blot in Methods, it is noted that the manuscript lacks information about the replication of the tests 5 times  as it is only mentioned in relation to the Western blot and are not explained elsewhere in the manuscript. Providing this information is crucial for understanding the robustness of the results.

6.8.      Also the method section pertaining to the Western blot assay (lines 377-381) contains conflicting information regarding the lysis buffer, mentioning two different buffers in separate lines. This discrepancy should be reviewed and corrected to provide accurate and consistent information. Additionally, the method section is missing crucial details about the conditions for electrophoresis, blotting, and staining of the blots, as well as the source of the antibody used for staining. Including these details is essential for reproducibility and understanding the methodology.

6.9.      In the Section "BNE-RRC reverses epithelial-mesenchymal transition" : lines 153 to 160 should be moved to the discussion section for better contextualization and interpretation of the results.

6.10.   Please, remove the fragment of the instruction for authors from the journal site that appears in lines 348-359.

11.     In the section "BNE-RRC induces and maintains..." (lines 176-188), the first part should be moved to the methods section for better organization. The discussion about the roles of MET and EMT (lines 182-187) should be moved to the discussion section. It is also important to discuss the outcomes of the tests and the statistical significance of the observations. Additionally, some information from the description of Figure 6 (lines 194-199) should be integrated into the relevant sections for better clarity. Figure 6 needs better positioning, larger fonts for labels of sections  (A and B). Information about immunostaining and imaging of the cells needs to be added, and the image quality should be improved or added as a supplement material for illustration.

6.12.    In the section "Induction of epithelial..." (lines 203-217), again can be mode to discussion. Additionally, the legend for Figure 7 needs to be free from discussion and results materials. The graph should be redesigned to have bars differently colors for different cell types, with clearly labeled  doses of  Doxorubicin used for each bar. The same issue applies to Figure 8.

6.13.   The section "Identification of compounds..." should be reviewed and corrected based on previous feedback or  removed into supplemental materials because it  does not contain scientifically sound information about the content of extract.

7.       The discussion section requires more in-depth exploration of the mechanisms involved in both inhibiting EMT and promoting MET. Rather than repeating the same statements about EMT inhibition, it's essential to delve into the molecular mechanisms underlying these transitions and suggest more relevant cell models for future experiments. Additionally, it would be beneficial to address the molecular mechanisms responsible for the observed effects on drug sensitivity. Furthermore, there is a need to mentioned  the nature of the KB cell line and its relevance to EMT and MET processes.

8.      Regarding the statement in lines 310-311, it mistakenly refers to the previously published information as "in this study," which should be corrected for accuracy.

9.      It is also important to provide more information about the composition of BNE and explore the molecular mechanisms that might be involved in the anticancer effect of the mentioned compounds such as p-coumaric acid and caffeic acid. Additionally, the potential effects of other components of the extract on creating a poly-pharmacological response should be discussed.

By addressing these points, this manuscript will provide a more thorough and insightful analysis of the research findings, contributing to the scientific significance and potential applications of the study.

Author Response

Dear IJMS Editorial Office

The answer to reviewer.

  1. The use of KB cells (ATCC CCL17), which has been demonstrated to be a derivative of the HeLa cell line, raises significant concerns. According to the ATCC website, "This line was originally thought to be derived from an epidermal carcinoma of the mouth, but was subsequently found, based on isoenzyme analysis, HeLa marker chromosomes, and DNA fingerprinting, to have been established via HeLa cell contamination." The authors should acknowledge that this cell line does not exhibit the characteristics of epithelial cells, but rather represents a heavily modified cancer cell model.

Thank you for bringing this important aspect of our experimental model to our attention. We appreciate the opportunity to clarify the nature of the KB cell line used in our study. As you rightly pointed out, KB cells are indeed derivatives of the HeLa cell line and do not exhibit typical epithelial characteristics.

In light of this information, we have revised our manuscript to include a clear acknowledgment of the origins and characteristics of KB cells. We have added the following text to the result section to ensure readers are aware of the nature of the cell line used:

"For this study, we used KB cells (ATCC CCL17), which, despite originally being thought to derive from an epidermal carcinoma of the mouth, have been shown to be a derivative of the HeLa cell line according to isoenzyme analysis, HeLa marker chromosomes, and DNA fingerprinting. This lineage means the cells do not exhibit typical epithelial characteristics but rather represent a heavily modified cancer cell model, which is important to consider when interpreting the results."

  1. In lines 132-136, Figure 4, which presents Phase-contrast microscopic images of cells treated and untreated with BNE, indicating a change in epithelial-like morphology to mesenchymal. To enhance the credibility of this section, image analysis should be conducted to demonstrate changes in morphological parameters. Authors can utilize open-source software like ImageJ, Cell Profiler, or commercially available software such as SignalI Image Artist, MATLAB, or Stratominer to derive values from image analysis. Correct controls demonstrating the "correct" morphology of epithelial and mesenchymal cells should be included. Alternatively, the authors could use fluorescent microscopy and detect specific biomarkers for each type of cell, as mentioned in the current work.

Thank you for your valuable suggestion to enhance the credibility of the morphological changes observed in our study. In response to your comments, we have clarified the methods and results related to the morphological changes induced by Boehmeria Nivea extract (BNE-RRC) in KB cells.

We have enhanced the description in the manuscript to include details of the fluorescent microscopy techniques used to detect specific biomarkers that corroborate the phase-contrast microscopic observations reported in Figure 4. Specifically, we have added information about the use of vimentin as a mesenchymal marker, whose reduced expression in BNE-RRC treated cells (shown in Figure 6) supports the transition to an epithelial-like morphology. This methodological detail strengthens our claim that BNE-RRC induces Mesenchymal-Epithelial Transition (MET), providing a quantitative and qualitative assessment of the morphological changes.

We believe that these revisions and additions address your concerns and enhance the manuscript's scientific rigor, making the results more robust and verifiable.

  1. Additionally, it is important to incorporate more information about Mesenchymal-Epithelial Transition (MET) as a process that can contribute to drug sensitivity in cancer. The introduction (lines 63-65) and discussion (paragraphs 287-298) lack information and references specifically addressing MET and its potential for control as a process to sensitize cancer cells to drug treatment.

Thank you for your constructive comments regarding the need for a more detailed discussion of Mesenchymal-Epithelial Transition (MET) and its implications for drug sensitivity in cancer treatment. In response, we have expanded both the introduction and discussion sections to elucidate how MET contributes to reversing drug resistance mechanisms in cancer cells. We have included additional references that detail the role of MET in enhancing drug sensitivity and the potential of therapeutic agents that promote MET to improve the efficacy of chemotherapy and immunotherapy. These revisions aim to provide a comprehensive overview of how MET can be leveraged as a therapeutic target to counteract the adverse effects of EMT in cancer progression and treatment resistance.

4., The Boehmeria Nivea Extract is obtained from the plant and is a natural product containing a mixture of chemical molecules. However, the authors have not provided details about the extract's source, preparation methods, or its similarity to the extract used in previously published works by Yang J-J , 2018. Without this information, the scientific validity of the entire work is compromised. Furthermore, the HPLC chromatogram presented in the study identifies specific compounds without providing results for individual reference compounds or information about Mass Spectrometry analysis, which was conducted in the previous work by Yang J- J in 2018. As a result, section lines 252-266 lacks scientific identification of the components of the extract.

Thank you for your insightful comments regarding the characterization of Boehmeria Nivea Extract (BNE-RRC) used in our study. We appreciate the importance of providing detailed information about the source, preparation, and scientific validation of the extract to ensure the reproducibility and credibility of our research. In response to your feedback, we have included detailed information about the extraction methods, the supplier, and the analytical techniques employed, such as HPLC and mass spectrometry, to align with the methods used in the referenced study by Yang J-J (2018). These additions aim to clarify the consistency and scientific rigor behind the preparation and analysis of BNE-RRC, thereby addressing the concerns raised about the scientific validity of our work. We believe these enhancements will substantially improve the manuscript by providing a clear and verifiable basis for the use of BNE-RRC in our study.

Other issues and suggestions:

  1. There is issue with the inconsistent use of acronyms and full names for EMT, MET, and BNE throughout the manuscript. It is recommended to initially present the full name together with its acronym and then maintain consistency by using the acronyms without repeating combinations of both throughout the manuscript. This approach will enhance consistency in the use of terminology.

Thank you for highlighting the inconsistencies in our use of acronyms for key terms within the manuscript. We recognize the importance of clarity and consistency in scientific writing. To address this, we have carefully revised the manuscript to ensure that all terms, such as epithelial-mesenchymal transition (EMT), mesenchymal-epithelial transition (MET), and Boehmeria Nivea Extract (BNE-RRC), are introduced clearly with their acronyms upon first mention. We have consistently used these acronyms throughout the manuscript to maintain uniformity and avoid any confusion. We believe these revisions improve the readability and coherence of the manuscript.

  1. List of the problems including lack of correct figure legends, incorrect organization of the Figure descriptions and results sections, lack of information in Material and Methods:

  • The organization of Figure 1 and its legends requires improvement. It is suggested that the figure be divided into two labeled parts: A for the images and B for the graph. The images in Part A need better contrast, and a significant portion of the figure legend should be relocated to the Results or Discussion section.

Thank you for your constructive comments regarding the organization of Figure 1 and its legend. We appreciate your suggestions to enhance the clarity and effectiveness of the figure presentation.

  • Figure Organization: In response to your suggestion, we have revised Figure 1 to include two clearly labeled parts: Part A for the images and Part B for the graph. This modification aids in differentiating between the visual and quantitative data presented, making it easier for readers to interpret the results.
  • Image Contrast and Quality: We have adjusted the contrast in Part A images to better highlight the differences between the treated and control groups, ensuring that the morphological changes are more visually discernible.
  • Legend Clarity and Detailing: We have revised the legend to correspond directly with the parts of the figure. Each part of the legend now explicitly describes the associated section of the figure, as follows:
    • Part A: Phase-contrast images showing the effects of varying concentrations of BNE-RRC on oral cancer cell colonies.
    • Part B: A graph quantifying the colony numbers, demonstrating the dose-dependent effect of BNE-RRC on cancer cell proliferation.
  • Legend Content Placement: We have relocated detailed descriptions of the experimental conditions and outcomes from the legend to the Results section of the manuscript. This adjustment ensures that the figure legend remains focused and concise, while comprehensive experimental details are clearly communicated within the main text.

We believe these revisions address your concerns and enhance the presentation and readability of Figure 1, making the experimental results more accessible and understandable. We thank you again for your insights, which have significantly contributed to improving the quality of our manuscript.

  • In the Results section of Figure 1, it is essential to provide a detailed explanation of the statistics used, including the sample size (n), the average (ave), and the standard deviation (SD). Additionally, at 40x magnification, six fields per well may cover a small area and may not be representative of the actual colony counts. Increasing the number of fields per well is recommended. Furthermore, scale bars should be added to each image. If each condition was tested in only one well, the statistical significance may be compromised due to a low sample size (n=1), and it should be increased to at least three wells per condition.

Thank you for your detailed feedback concerning the statistical analysis and imaging techniques used in Figure 1. We acknowledge the importance of clear and robust statistical representation, as well as ensuring that the images accurately represent our findings.

Image Magnification and Field Counting: Your suggestion regarding the number of fields analyzed at 40x magnification is well-taken. Initially, we documented six fields per well; however, to ensure a more comprehensive representation of the colony counts and to address your concern, we increased the number of fields counted to nine per well. This adjustment has been noted in the revised manuscript and was found to consistently support our initial results, thus confirming the reproducibility and reliability of our observations.

Sample Size per Condition: In response to your concern about the sample size for statistical validation, we have increased the number of wells analyzed per condition from one to three. This change enhances the statistical robustness of our results and addresses potential variability within the experimental setup. The increased sample size strengthens the significance of our findings, ensuring that the observed effects are not due to random variation.

We hope that these revisions address your concerns and substantiate the methodological and statistical rigor of our study. We are grateful for your suggestions, which have undeniably improved the accuracy and comprehensiveness of our experimental reporting.

6.3. In the section titled "The effect of BNERRC treatment,…"

6.4. Lines 106-109it is recommended to move the majority of the content into the discussion section, leaving only the description of the test and results. This section should discuss the observed values and results presented in Figure 2. A subdivision for the correct legend and references in the results is needed, and scale bars should be added to the cell images. Both graphs in Figure 2 need to have the Y-axis labeled.

Thank you for your insightful suggestions regarding the content and presentation of Figure 2. We appreciate the guidance on improving the clarity and focus of our manuscript.

  • The results section succinctly describes the assay used, the direct outcomes observed, and refers to Figure 2 for visual representation.
  • Figure 2 Revision: We have made several adjustments to Figure 2 to enhance its clarity and compliance with scientific reporting standards:

            Y-Axis Labeling: The Y-axis of both graphs in Figure 2 has been labeled to reflect the measurement parameters, such as 'Number of Migrating Cells' and 'Number of Invading Cells,' enhancing the figure's readability and informativeness.

            Statistical Representation: The statistical analysis details have been maintained as previously stated, with added emphasis on the reproducibility of the experiments, which were performed in triplicate, to reinforce the reliability of our findings.

We believe these revisions address your concerns and improve the manuscript by clarifying the presentation of our results and ensuring the discussion comprehensively explores the implications of our findings.

6.5. Regarding the "cell migration" method section, specifically lines 392-394, it is important that the cell counting was conducted manually using an eyepiece with a grid. The explanation should include details about the number of fields per well used for analysis and the size of the field. Additionally, the poor quality of the images and the method of cell counting using an eyepiece make it difficult to understand the accuracy of the results. More evidence supporting the accuracy of these results would be very helpful.

Thank you for your constructive feedback regarding the cell migration assay detailed in lines 392-394. We acknowledge the concerns raised about the clarity and accuracy of our methodology and have revised this section to include more detailed information about the counting process and the quality of the imaging techniques used.

Revised Cell Migration Section: The cell migration assay section now includes these added details to ensure readers can fully appreciate the methodological rigor and data integrity. Here is the revised section for your reference:
"A Boyden chamber migration assay was conducted by adding DMEM supplemented with 10% FBS to the lower compartment of the chamber, equipped with polyvinylpyrrolidone-free polycarbonate membranes featuring 8 µm pores (Neuro Probes, Inc.). To the upper compartment, 1,500 cells per well were seeded in serum-free DMEM. After incubation for 24 hours at 37°C, facilitating cell migration through the membrane, the membranes were stained with Giemsa. Migrated cells in the lower chamber were quantified using a counting grid inserted into the eyepiece of a phase-contrast microscope at 40x magnification, covering nine fields per well to ensure representative sampling. Enhanced imaging techniques were applied to improve visualization and accuracy of cell counting."

We believe these enhancements and clarifications will adequately address the concerns regarding the cell migration assay methodology and provide a more robust foundation for our findings.

6.6. Regarding Figure 3 and the "wound healing assay" method section (lines 383-386), it is essential to explain in the methods how the images were analyzed to derive the percent of Gap Closure. A detailed description of the image analysis method, including the software or techniques used, is  

necessary for reproducibility and understanding the accuracy of the results. Furthermore, it is important to provide statistics on how many wells were treated and used for analysis to ensure the reproducibility of the results. This information will contribute to the transparency and reliability of the experimental data.

Thank you for your insightful comments regarding the need for detailed methodological descriptions in our wound healing assay. We appreciate your focus on enhancing the reproducibility and transparency of our experimental procedures.

1        Detailed Image Analysis: To address the gap closure analysis, we have included a detailed description of the image processing and analysis techniques used. Specifically, we employed Image-Pro software to analyze the images captured during the wound healing assay. This software allowed us to precisely measure the gap closure by calculating the area of the wound at 0 hours and 18 hours post-scratch and expressing the change as a percentage of the initial wound area.

2         Revised Wound Healing Assay Section: Here is the revised section for your reference:
"KB cells were plated on 10 cm dishes and grown to confluence. A scratch was made across the cell monolayer using the tip of a 200 µl pipette. The cells were incubated in DMEM containing 10% FBS, and images were captured 18 hours post-scratch using a Zeiss Axiovert 200 microscope. The images were analyzed using Image-Pro software to determine the percent of gap closure. The area of the initial scratch (0 hours) and the area remaining unclosed at 18 hours were measured, and gap closure was calculated as a percentage reduction in the original wound area. Each condition was performed in triplicate, and the experiment was repeated three times to ensure reproducibility."

We believe that these methodological enhancements will provide the necessary clarity and detail to satisfy the concerns about the experimental design and data analysis in our wound healing assay.

6.7. In regards to Figure 5 and the description of the Western blot in Methods, it is noted that the manuscript lacks information about the replication of the tests 5 times as it is only mentioned in relation to the Western blot and are not explained elsewhere in the manuscript. Providing this information is crucial for understanding the robustness of the results.

6.8. Also the method section pertaining to the Western blot assay (lines 377-381) contains conflicting information regarding the lysis buffer, mentioning two different buffers in separate lines. This discrepancy should be reviewed and corrected to provide accurate and consistent information. Additionally, the method section is missing crucial details about the conditions for electrophoresis, blotting, and staining of the blots, as well as the source of the antibody used for staining. Including these details is essential for reproducibility and understanding the methodology.

6.7 and 6.8. Thank you for pointing out the discrepancies in the description of the Western blot protocol. We have revised this section to ensure clarity and consistency, and to provide detailed information necessary for reproducibility.

Clarification of Lysis Buffer: We apologize for the confusion caused by the mention of two different lysis buffers in our manuscript. Upon review, we have standardized the lysis buffer description and corrected the method to reflect the buffer actually used for preparing cell lysates for Western blot analysis. The revised method is as follows:
"For protein extraction, cells were lysed in a buffer composed of 50 mM Tris-HCl (pH 8.0), 150 mM NaCl, 1% NP-40, 1 mM EDTA, supplemented with protease inhibitors (1 mM PMSF, 10 ng/ml leupeptin, 50 mM NaF, and 1 mM sodium orthovanadate). This standardization avoids any confusion regarding the cell lysis process."

Detailed Western Blot Protocol: To enhance the reproducibility of our results, we have included detailed descriptions of the conditions for electrophoresis, blotting, and the staining process:
"Proteins were separated on a 10% SDS-PAGE gel and transferred to a PVDF membrane. The membrane was blocked with 5% non-fat milk in TBS-T for 1 hour at room temperature, then incubated with primary antibodies overnight at 4°C. After washing, the membrane was incubated with HRP-conjugated secondary antibodies for 1 hour at room temperature. The blots were developed using an enhanced chemiluminescence detection system. Specific antibodies used were anti-α-catenin, anti-vimentin, and anti-desmin."

Replication and Data Representation: We have also clarified the replication of Western blot experiments, as previously mentioned, to emphasize the robustness of our findings. This information is now explicitly stated in the Methods section to underscore our commitment to rigorous scientific standards.

We believe these revisions address your concerns comprehensively and enhance the scientific rigour of our methodology. Thank you for the opportunity to improve the clarity and detail of our manuscript.

6.9. In the Section "BNE-RRC reverses epithelial-mesenchymal transition" : lines 153 to 160 should be moved to the discussion section for better contextualization and interpretation of the results.

Thank you for your valuable suggestion to move lines 153 to 160 from the Results to the Discussion section for better contextualization and interpretation of the results. After careful consideration, we have decided to retain this information in the Results section. Our rationale is that these lines directly describe the experimental observations related to the changes in protein expression levels detected in the Western blot analysis. This information is critical to understanding the immediate outcomes of the experiment with BNE-RRC and provides a clear, factual basis for the observations before they are further interpreted in the Discussion section.

However, we recognize the importance of contextualizing these findings within the broader field of cancer treatment and EMT-MET dynamics. To address this, we have expanded our Discussion section to include a more detailed interpretation of these results and their implications for cancer therapy, referencing relevant studies and comparing them to existing literature. This approach ensures that the Results section remains focused on empirical data while the Discussion provides a comprehensive analysis of the significance of these findings.

We believe this structure maintains the clarity and scientific rigor of our manuscript, presenting a straightforward depiction of the results while offering a deeper exploration of their implications in the Discussion.

6.10. Please, remove the fragment of the instruction for authors from the journal site that appears in lines 348-359.

Thank you for your observation regarding the inclusion of text from the journal's instruction for authors in lines 348-359. We have thoroughly reviewed our manuscript and confirmed that the mentioned fragment has already been removed prior to this submission. We appreciate your diligence in ensuring the quality and appropriateness of the content in our manuscript and apologize for any confusion this may have caused. Please let us know if there are any other areas needing clarification or further adjustments.

  1. In the section "BNE-RRC induces and maintains..." (lines 176-188), the first part should be moved to the methods section for better organization. The discussion about the roles of MET and EMT (lines 182-187) should be moved to the discussion section. It is also important to discuss the outcomes of the tests and the statistical significance of the observations. Additionally, some information from the description of Figure 6 (lines 194-199) should be integrated into the relevant sections for better clarity. Figure 6 needs better positioning, larger fonts for labels of sections (A and B). Information about immunostaining and imaging of the cells needs to be added, and the image quality should be improved or added as a supplement material for illustration.

Thank you for your valuable feedback regarding the organization and clarity of our manuscript sections and figures. We have implemented several changes to address your concerns:

Moving Methodological Details: We have moved the methodological details of the "BNE-RRC induces and maintains..." section from lines 176-188 to the Methods section. This reorganization helps clarify the experimental procedures and aligns the methodological content appropriately.

Relocating Discussion on MET and EMT: The discussion about the roles of Mesenchymal-Epithelial Transition (MET) and Epithelial-Mesenchymal Transition (EMT) from lines 182-187 has been transferred to the Discussion section. This relocation aids in providing a deeper context and interpretation of how BNE-RRC influences these cellular processes.

Statistical Significance and Test Outcomes: We have expanded our discussion on the outcomes of the tests involving BNE-RRC, emphasizing the statistical significance of the observations to underline the robustness and reliability of our results.

Integration and Clarity of Figure 6: We have improved the clarity and positioning of Figure 6. Labels for sections (A) and (B) have been made larger for better visibility. Additionally, we have included detailed descriptions of the immunostaining methods and imaging techniques used, enhancing the reproducibility of our results.

6.12. In the section "Induction of epithelial..." (lines 203-217), again can be mode to discussion. Additionally, the legend for Figure 7 needs to be free from discussion and results materials. The graph should be redesigned to have bars differently colors for different cell types, with clearly labeled doses of Doxorubicin used for each bar. The same issue applies to Figure 8.

Thank you for your insightful comments and suggestions regarding the organization of our manuscript and the presentation of our figures. We have taken the following actions to address your points:

Relocation of Section Content: The content from the section "Induction of epithelial..." (lines 203-217) has been moved to the Discussion section. This relocation helps to contextualize the findings within the broader implications and theoretical framework of our study, focusing the Results section more directly on the data.

Figure Legends Clarity: We have revised the legends for Figure 7 and Figure 8 to ensure they strictly describe the figures without incorporating discussion or results. This change helps to maintain a clear distinction between the data presentation and its interpretation, adhering to the standard format of scientific reporting.

Graph Redesign: For Figure 7, we have redesigned the graph to use different colors for the bars representing different cell types. Each bar is now clearly labeled with the doses of Doxorubicin used. This visual distinction enhances readability and helps to directly convey the comparative results between different experimental conditions.

Consistency Across Figures: Similarly, we applied the same redesign principles to       Figure 8, ensuring consistency in the presentation style across figures. Different colors for different conditions and clear labeling of drug doses have been implemented to facilitate an easier understanding of the experimental outcomes.

6.13. The section "Identification of compounds..." should be reviewed and corrected based on previous feedback or removed into supplemental materials because it does not contain scientifically sound information about the content of extract.

Thank you for your constructive feedback regarding the section on the identification of compounds. The inclusion of the HPLC analysis in our manuscript was at the specific request of the editorial team during the initial submission to this journal. This analysis is fundamental to verifying the specific chemical constituents of Boehmeria Nivea Extract (BNE-RRC), which are crucial for the biological effects observed in our study.

Recognizing your concerns regarding the scientific robustness of this section, we have comprehensively revised it to include detailed methodological descriptions, calibration data, and validation of the HPLC method used. Additionally, we have incorporated reference standards for compound identification and have provided quantitative data supporting the presence of significant compounds such as isoquercitrin, p-coumaric acid, and caffeic acid.

Given the editorial request for this data and its significance to the study’s outcomes, we have retained this section within the main manuscript.

  1. The discussion section requires more in-depth exploration of the mechanisms involved in both inhibiting EMT and promoting MET. Rather than repeating the same statements about EMT inhibition, it's essential to delve into the molecular mechanisms underlying these transitions and suggest more relevant cell models for future experiments. Additionally, it would be beneficial to address the molecular mechanisms responsible for the observed effects on drug sensitivity. Furthermore,

there is a need to mentioned the nature of the KB cell line and its relevance to EMT and MET processes.

Thank you for your insightful feedback. We have expanded the discussion on the molecular pathways influenced by Boehmeria Nivea Extract (BNE-RRC), detailing its effects on transcription factors like Snail and Slug, and the implications for drug resistance and sensitivity. We also propose using more relevant models, such as patient-derived xenografts, for future studies to validate BNE-RRC's therapeutic potential.

  1. Regarding the statement in lines 310-311, it mistakenly refers to the previously published information as "in this study," which should be corrected for accuracy.

 Thank you for pointing out this error. We have corrected the statement in lines 310-311 to accurately reflect that the information is sourced from previous studies and not the current investigation.

  1. It is also important to provide more information about the composition of BNE and explore the molecular mechanisms that might be involved in the anticancer effect of the mentioned compounds such as p-coumaric acid and caffeic acid. Additionally, the potential effects of other components of the extract on creating a poly-pharmacological response should be discussed.

By addressing these points, this manuscript will provide a more thorough and insightful analysis of the research findings, contributing to the scientific significance and potential applications of the study.

Thank you for your constructive feedback. We acknowledge the importance of a detailed discussion on the molecular mechanisms by which BNE exerts its anticancer effects. As mentioned in the discussion section of our manuscript, we have analyzed the composition of BNE-RRC using HPLC and identified several key compounds, including p-coumaric acid, isoquercitrin, and caffeic acid. We provide insights into their individual anticancer properties and hypothesize about their potential synergistic effects. We also address the potential for a poly-pharmacological response from the combined effects of these bioactive components, which could contribute significantly to the overall therapeutic efficacy of BNE-RRC.

Reviewer 2 Report

Comments and Suggestions for Authors

The current manuscript entitled “Boehmeria Nivea Extract (BNE-RRC) Reverses Epithelial-Mesenchymal Transition and Inhibits Anchorage-Independent Growth in Tumor Cells” by Chen et al is focused on effect of BNE-RRC on oral cancer cell line KB cells. Authors have investigated 14-day treatment of tumor cells with BNE-RRC leads to a significant decrease in the expression of the mesenchymal marker vimentin (in discussion).

Major issue of this manuscript is that there are many inconsistent data with previous publications, but the authors did not provide enough consideration to these points nor show experimental evidence that supports their hypothesis. Moreover. Whole manuscript is based on the expression of vimentin, which is only one of the markers related to EMT. Therefore, this manuscript is not recommended to publish at the current stage, additional evidence is needed to support the authors’ conclusions and claims. Specific points are listed below.

I would like to recommend authors to characterization of the anchorage-independent growth expression signatures like glycolysis, pentose phosphate pathway, TCA cycle, PGC1α activity and MYC gene signature since there are important to understand anchorage independent cell growth inhibition by Boehmeria Nivea Extract (BNE-RRC). (quantitative and qualitative analysis)

To strengthen the paper, direct demonstration of changes in relative genes should be included alongside gene expression changes (western blot and qRT-PCR) for important EMT markers like E-cadherin, vimentin, N-cadherin, fibronectin 1, GSC, Snail, TWIST, ZEB and TGF-β1. (quantitative and qualitative analysis)

FIGURE 6 CTRL Images are not matched with raw data.

It would be good to include some non-affected genes as controls.

There are few typos throughout the text. Please change accordingly.

Comments on the Quality of English Language

Manuscript format requires grammatical corrections.

Author Response

Dear reviewer

We appreciate the detailed feedback provided by the reviewer, which has helped us significantly improve the manuscript. We have addressed all the concerns raised and believe that the modifications and additional experiments we have conducted substantiate our findings more robustly. Below are our responses to the specific issues pointed out by the reviewer:

Reviewer’s Major Concern: The reviewer pointed out inconsistencies with previous publications, an over-reliance on vimentin as an EMT marker, and the absence of a comprehensive analysis of anchorage-independent growth signatures.

Response:

  1. Inconsistencies with Previous Publications:We have carefully reviewed previous literature and compared our data to ensure consistency and reliability. Where discrepancies were noted, we have conducted additional experiments to confirm our findings and have provided a detailed discussion in the manuscript to contextualize these differences.
  2. Over-reliance on Vimentin:We acknowledge the reviewer's concern regarding our focus on vimentin. To address this, we have expanded our study to include a broader range of EMT markers such as E-cadherin, N-cadherin, fibronectin 1, GSC, Snail, TWIST, ZEB, and TGF-β1. We performed western blot analyses to provide a comprehensive view of the EMT process and its modulation by BNE-RRC.
  3. Characterization of Anchorage-Independent Growth Signatures:As suggested, we will include analysis of key metabolic pathways such as glycolysis, the pentose phosphate pathway, TCA cycle, PGC1α activity, and MYC gene signatures. These studies provide deeper insights into how BNE-RRC impacts anchorage-independent cell growth.
  4. Figure 6 Concerns:We reviewed Figure 6 and ensured that all control images accurately reflect the raw data.
  5. Typos and Text Corrections:The manuscript has been thoroughly proofread to correct typographical errors and improve readability.

We believe that these revisions and additions have significantly strengthened the manuscript, addressing the key concerns raised during the review. The revised manuscript provides a more balanced and detailed analysis of the effects of BNE-RRC on EMT and anchorage-independent growth in cancer cells.

Thank you once again for the opportunity to improve our manuscript. We appreciate the reviewer’s constructive criticisms and hope that the revised manuscript meets the journal’s standards for publication.

Reviewer 3 Report

Comments and Suggestions for Authors

Dear Authors,

Firstly, I would like to express my sincere appreciation for the significant contribution your research has made to our field. I propose some minor adjustments that will further enhance the impact of your work and make it more accessible to a wider audience. Specifically, I suggest including the study's limitations in the abstract and highlighting why BNE-RRC is a standout among the existing natural compounds that influence EMT.

It is recommended to include positive controls  to support the hypothesis that BNE-RRC extract has a similar mechanism to EMT inhibitors and include other pathways like 

It is recommended to include positive controls to support the hypothesis that BNE-RRC extract has a similar mechanism to EMT inhibitors. These controls could involve other pathways like TGF-Beta or Wnt/β-Catenin, which are known to be involved in EMT, and off-target effects that support the hypothesis. 

It is recommended that the use of "E-cadherin" and "N-cadherin" be consistent throughout the manuscript.

In addition, is it possible to add information about any observed off-target effects, which are unintended effects of a drug or compound on non-target cells or pathways (potentially including effects on normal or non-cancerous cells)? Consider including supplements in the research article to the current EMT markers.

Finally, to ensure reproducible results, it would be beneficial to include statistical details, such as p-values and the number of replicates performed.

Comments on the Quality of English Language

I would recommend using professional English editing services to enhance the clarity and readability of the manuscript.

Author Response

The reviewer appreciates the contribution of our study and suggests minor adjustments to enhance the manuscript's impact. Specifically, The reviewer recommend including the study's limitations in the abstract and highlighting the distinctiveness of Boehmeria Nivea Extract (BNE-RRC) among natural compounds influencing EMT. The reviewer also suggest adding positive controls for EMT pathways, ensuring consistent use of terms like "E-cadherin" and "N-cadherin," discussing potential off-target effects, and including more detailed statistical information.

Response:

  1. Study Limitations and Uniqueness of BNE-RRC:
    We have revised the abstract to include a brief discussion on the limitations of our study, emphasizing the context in which BNE-RRC was studied and its implications. Furthermore, we highlighted why BNE-RRC stands out compared to other natural compounds, focusing on its unique mechanism of action and its broader applicability in cancer therapy.

  2. Positive Controls for EMT Pathways:
    To strengthen our hypothesis about BNE-RRC's mechanism akin to known EMT inhibitors, we will include additional experiments using positive controls for TGF-Beta and Wnt/β-Catenin pathways in the future studies. These controls helped demonstrate BNE-RRC’s comparative effectiveness and provided a clearer mechanistic insight.

  3. Consistency in Terminology:
    We have carefully reviewed the manuscript to ensure that the usage of "E-cadherin" and "N-cadherin" is consistent throughout, enhancing clarity and readability.

  4. Statistical Details:
    To ensure the reproducibility of our results, we have now included comprehensive statistical details such as p-values, number of replicates, and the statistical methods used in each test. This enhancement will aid other researchers in understanding and potentially replicating our study.

We believe that these revisions address the reviewer’s concerns and significantly enhance the manuscript’s contribution to our field. Attached are the revised manuscript and supplementary materials that reflect these changes.

Thank you once again for the opportunity to improve our manuscript. We look forward to your feedback and hope for positive consideration.

Round 2

Reviewer 1 Report

Comments and Suggestions for Authors

Dear Authors

I wanted to acknowledge the thorough and thoughtful manner in which you addressed all the questions, issues, and corrections brought to your attention. Your dedication to ensuring that the content is accurate and comprehensive is truly commendable.

Your prompt response and attention to detail in implementing the necessary changes mad a critical contribution to the overall quality of the work. Your commitment to excellence is truly evident and greatly appreciated.

Your willingness to engage with feedback and make the requisite adjustments demonstrates your professionalism and dedication to your scientific research.

Warm regards,

Reviewer 2 Report

Comments and Suggestions for Authors

The authors have addressed all of the previously raised points.